# Optimizing Well Placement for Sustainable Irrigation: A Two-Stage Stochastic Mixed Integer Programming Approach

Wanru Li [1,*], Mekuanent Muluneh Finsa [2,3], Kathryn Blackmond Laskey [1], Paul Houser [4], Rupert Douglas-Bate [5] and Kryštof Verner [6,7]

1. Department of Systems Engineering and Operations Research, George Mason University, Fairfax, VA 22030, USA; klaskey@gmu.edu
2. Institute of Hydrogeology, Engineering Geology and Applied Geophysics, Charles University, 128 44 Prague, Czech Republic; finsam@natur.cuni.cz or mekuanent.muluneh@amu.edu.et
3. Water Resource Research Center, Arba Minch University, Arba Minch P.O. Box 21, Ethiopia
4. Department of Geography and Geoinformation Science, George Mason University, Fairfax, VA 22030, USA; phouser@gmu.edu
5. Global MapAid, Aylesbury HP17 8RZ, UK; rupertdouglasbate@globalmapaid.org
6. Czech Geological Survey, Klárov 3, 118 21 Prague, Czech Republic; krystof.verner@geology.cz
7. Institute of Petrology and Structural Geology, Faculty of Science, Charles University, Albertov 6, Prague 2, 128 00 Prague, Czech Republic
* Correspondence: wli15@gmu.edu

**Abstract:** Utilizing groundwater offers a promising solution to alleviate water stress in Ethiopia, providing a dependable and sustainable water source, particularly in regions with limited or unreliable surface water availability. However, effective decision-making regarding well drilling and placement is essential to maximize groundwater resource potential, enhancing agricultural productivity, reducing hunger, and bolstering food security in Ethiopia. This study concentrates on the development of two-stage stochastic mixed integer programming (SMIP) models to optimize well placement for sustainable agricultural irrigation, considering uncertain demand scenarios. Additionally, a deterministic mixed integer programming model is formulated for comparison with the two-stage SMIP. Experiments are conducted to explore various demand scenario distributions, revealing that the optimized total cost for the two-stage SMIP generally exceeds that of a deterministic setting, aligning with the two-stage SMIP's focus on long-term benefits. Moreover, slight differences are observed in well layouts under different assumption scenarios. The study also examines the impact of selected parameters, such as fixed construction costs, per-meter drilling costs, and demand scenarios. The out-of-sample performance shows that the stochastic model is more flexible and resilient, with 11% and 4% lower costs than deterministic cases 1 and 3, respectively. This flexibility provides a more robust long-term strategy for well placement and resource allocation in groundwater management.

**Keywords:** well placement optimization; well layout optimization; mixed integer programming (MIP); two-stage stochastic mixed integer programming (SMIP); Bilate watershed; southern Ethiopia; sustainable irrigation

## 1. Introduction

Ethiopia is characterized by a complex geological structure and rugged topography, with extremely diverse hydrogeological conditions. These include (a) Neoproterozoic, low- to high-grade metamorphic rocks that are part of the Arabian-Nubian Shield, overlain by (b) platform sedimentary units ranging from the Lower Paleozoic to the Paleogene, as seen in the large Ogaden Basin, Blue Nile Basin, and Mekele Basin, and (c) Eocene to Holocene volcanic rocks and volcaniclastic deposits associated with the active NNE-SSV trending Main Ethiopian Rift [1,2]. Especially in light of these specific natural factors, Ethiopia faces a pressing challenge in ensuring access to clean and sustainable water

sources for its burgeoning population. With over 60 million people lacking access to clean water, Ethiopia ranks among the countries with the lowest water supply coverage in sub-Saharan Africa [3]. Compounded by climate variability, recurrent droughts, and geological risks associated with slope deformations, the country's water resources are under significant strain, threatening the livelihoods and well-being of millions of Ethiopians [4]. Inadequate infrastructure and poor water management exacerbate the situation, further deepening the water crisis in both rural and urban areas [5]. Utilizing groundwater presents a promising avenue for mitigating water stress and ensuring sustainable water access in Ethiopia. Groundwater resources, if managed effectively, offer a reliable and resilient water source, particularly in regions where surface water availability is limited or erratic [6]. However, unlocking the potential of groundwater requires strategic decision-making in well drilling and placement.

Well drilling decision-making holds significant importance in optimizing groundwater utilization for sustainable development in Ethiopia. The strategic placement of wells with regard to hydrogeological conditions can enhance water access, improve agricultural productivity, and support rural livelihoods. However, managing groundwater resources involves uncertainties, particularly regarding future demand, recharge rates, and water table depth. Often, these characteristics only become fully known after drilling has occurred, requiring adaptive risk management strategies. By continuously adjusting management practices as new information becomes available, we can ensure more effective, long-term sustainability. Optimizing well placement and layout not only minimizes water wastage, reduces energy consumption, and mitigates the risk of groundwater depletion but also allows for flexible responses to these uncertainties. This adaptive, holistic approach to groundwater management benefits farming communities while contributing to broader societal goals such as poverty reduction, food security, and environmental sustainability [7].

There are two main optimization techniques that have been applied in well placement problems: conventional optimization methods and non-conventional methods. Conventional optimization methods have been extensively employed for well placement optimization in utilizing groundwater for irrigation. Drawing insights from a previous study [8], conventional approaches often utilize mathematical programming techniques such as linear programming (LP), nonlinear programming (NLP), integer programming (IP), mixed-integer programming (MIP), stochastic mixed integer programming (SMIP), and adjoint methods. In 2014, Tafteh et al. formulated a LP model for the optimization of irrigation water distribution [9]. Studies by authors such as Ma et al. (2019) [10] and Kuvichko and Ermolaev (2020) [11] have demonstrated the efficacy of MIP in optimizing well locations. Also, research by Liu et al. (2015) has optimized the layout of pumping wells based on NLP models for irrigation planning and management [12]. Additionally, Halilovic et al. (2022) introduced a new gradient-based optimization method using the adjoint approach for well layouts in groundwater heat pumps, facilitating continuous well locations and accommodating a large number of wells. Their method demonstrates versatility in optimizing layouts for multiple neighboring or single systems, as evidenced by a real case study involving 10 systems, showcasing its practical applicability [13]. Furthermore, the study by Bayer et al. in 2008 introduced a novel approach to solving stochastic optimization problems with multiple equally probable realizations of uncertain parameters, utilizing dynamically ordered stacks of realizations during the search process. By applying this technique to a water supply well field design problem, the study demonstrates that simple stack ordering can significantly reduce computational effort by up to 97% without compromising optimization results, offering promise for similar water management and reliability-based design challenges [14]. However, despite their ability to ensure optimality under certain conditions, conventional methods may encounter challenges, such as getting trapped in local optima.

In contrast, gradient-free optimization methods as non-conventional methods present promising alternatives for well placement optimization in irrigation systems, offering robustness against complex objective functions and solution spaces. Researchers such

as Emerick et al. (2009) implemented a genetic algorithm to optimize the number and location of the wells [15]. Investigations by researchers such as Sharifipour et al. (2021) have explored the application of the shuffled frog leaping algorithm, particle swarm optimization, and genetic algorithm in optimizing well placement problems. They found that the shuffled frog leaping algorithm outperformed other algorithms, demonstrating superior objective function values and well spacing in both intermediate and late optimization stages [16]. These gradient-free optimization techniques, as highlighted in [8], do not rely on gradient information and employ diverse search strategies to explore solution spaces iteratively. While they provide flexibility and scalability, a notable disadvantage of gradient-free optimization methods is their potential for slower convergence compared to gradient-based methods, which can result in longer computational times, particularly for high-dimensional optimization problems and large-scale irrigation systems. AlQahtani et al. (2013) conducted a comparison between evolutionary metaheuristics and mathematical optimization methods for the well placement problem. Their study revealed that mathematical optimization outperformed genetic algorithms by yielding superior solutions in a shorter computational timeframe [17].

Many previous research studies have investigated the optimization of pumping well layouts to enhance groundwater extraction efficiency. However, some of the studies focus on deterministic models that may not adequately account for the uncertain water demand and the need for sustainable groundwater development [9–12,18]. Additionally, some papers have relied on simplified models that assume an infinite aquifer, overlooking the potential risks of groundwater depletion and adverse environmental impacts [10,12]. To ensure the sustainable development of groundwater resources, it is essential to incorporate groundwater level and recharge parameters into the optimization model. By considering the variability of groundwater levels and recharge rates over different spaces, as well as the finite nature of aquifers, these parameters can provide valuable insights into the long-term viability of well placement strategies. Moreover, integrating sustainability constraints into the optimization framework can help balance water extraction with aquifer replenishment, thus safeguarding groundwater resources for future generations.

In this study, we chose to use mixed-integer programming (MIP) and two-stage stochastic mixed-integer programming (SMIP) over non-conventional methods such as genetic algorithms or shuffled frog leaping. The primary reason for this choice is the deterministic nature of MIP and its proven effectiveness in handling well placement optimization problems with precise constraints and objectives. Additionally, two-stage SMIP allows us to incorporate uncertainty in demand scenarios, providing a more robust solution under varying conditions. While non-conventional methods like genetic algorithms and shuffled frog leaping offer flexibility and robustness, they often require more computational effort and may not guarantee optimal solutions. Our initial focus on MIP and two-stage SMIP was due to their structured approach and computational efficiency, making them suitable for the preliminary phase of this research.

This study aims to model and optimize optimal drilling locations to minimize costs while meeting farm demand and ensuring sustainable groundwater development for subsistence agriculture in water-scarce regions. To achieve this objective, MIP and two-stage SMIP models are constructed. The two-stage stochastic model offers the benefit of long-term planning by considering uncertain demand scenarios. Therefore, our focus in this study is on the two-stage SMIP. These models are developed using predicted groundwater levels [19] and estimated groundwater recharge [20] from our previous studies conducted in a larger region that encompasses the current study area. The specific study area selected for analysis is a small region in the western flank of the Bilate watershed (westernmost part of the Sidama Region) in southern Ethiopia. This study also explores various uncertain demand scenarios for the two-stage SMIP by employing different distributions and comparing the outcomes. Additionally, we present the well layout maps for each experiment and analyze the impact of key parameters.

The remainder of this paper is organized as follows: Section 2 delineates the problem formulation for both the deterministic and stochastic models and provides an overview of the study area and the data utilized; Section 3 offers a comprehensive presentation of the main results from the optimization models and shows optimal well layouts; Section 4 discusses the impact of different parameters, provides an analysis on out-of-sample performance, discusses the limitations of this study, and suggests potential avenues for future research; finally, Section 5 concludes the paper and summarizes key findings.

## 2. Materials and Methods

### *2.1. Problem Formulation and Reformulation*

In this section, we provide problem formulation and reformulation for the deterministic MIP and two-stage SMIP in Sections 2.1.1 and 2.1.2, respectively.

2.1.1. Deterministic Mixed Integer Programming Model

Mixed-integer programming (MIP) is an optimization technique that involves decision variables that can be both continuous and discrete, allowing for complex modeling of real-world problems. It is widely used in various fields to find optimal solutions under given constraints, such as scheduling [21,22] and portfolio optimization [23,24]. In this section, we will introduce the sets, parameters, decision variables, formulation, and reformulation of the deterministic MIP.

*Sets:*

- *I*: set of farm sites, indexed by *i*.
- *J*: set of potential well sites, indexed by *j*.

*Parameters:*

- $c_j$: construction fixed cost at site *j*.
- $u_j$: unit cost for construction at site *j*.
- $t_{ij}$: transportation cost from *j* to *i*.
- $d_i$: demand at farm *i*.
- $l_{swl_j}$: static water level of the well at site *j*.
- **r**: average recharge quantity.
- **s**: area of the study region.
- $w_{max}$: the deepest depth the well can be drilled.
- $l_{min}$: the minimum difference between the depth well $w_j$ and static water level $l_{swl_j}$.

*Decision variables:*

- $x_j$: binary, equal to 1 if the well at location j is placed, 0 otherwise.
- $y_{ij}$: quantity of water that is transported from site *j* to farm *i*.
- $w_j$: depth of the well to drill at site *j*.
- $K_j$: capacity of the well at site *j*.

*Deterministic MIP problem formulation:*

Our objective is to minimize the summation of the total construction cost, the total transportation cost, and the cost for drilling different depths of wells. The objective function is determined in collaboration with experts in the Arba Minch Water Technology Institute. The objective function is defined as:

$$\min_x \sum_{j=1}^{J} c_j x_j + \sum_{i=1}^{I} \sum_{j=1}^{J} t_{ij} y_{ij} + \sum_{j=1}^{J} u_j w_j, \tag{1}$$

Constraint (2) ensures that all water demand from the farms is met. Constraint (3) requires that each constructed well *j* must have a capacity $K_j$. The total quantity of water transported from well *j* to all the farms must be less than or equal to $K_j$, the capacity of

that specific well. Constraint (4) specifies that the capacity $K_j$ is defined as the volume of groundwater above the deepest point of the well. Figure 1 presents a conceptual plot illustrating the capacity of a well. Constraint (5) mandates that the total amount of pumped-out water must be less than the quantity of recharged water. Constraint (6) requires that each constructed well j must have a depth less than the maximum allowable depth. Constraint (7) indicates that the lower bound of the well depth $w_j$ should be greater than the static water level plus the minimum difference between the well depth and the static water level. Constraints (8) and (9) are standard binary and non-negative constraints.

$$s.t. \qquad \sum_{j=1}^{J} y_{ij} = d_i, \quad \forall i \qquad (2)$$

$$\sum_{j=1}^{J} y_{ij} \le K_j x_j, \quad \forall i \qquad (3)$$

$$K_j = s * \left( w_j - l_{swl_j} \right), \quad \forall j \qquad (4)$$

$$\sum_{i=1}^{I} \sum_{j=1}^{J} y_{ij} \le r \qquad (5)$$

$$w_j \le w_{max}, \quad \forall j \qquad (6)$$

$$w_j \ge \left( l_{swl_j} + l_{min} \right) x_j, \quad \forall j \qquad (7)$$

$$y_{ij} \ge 0, \quad \forall i, \forall j \qquad (8)$$

$$x_j \in \{0, 1\}, \quad \forall j \qquad (9)$$

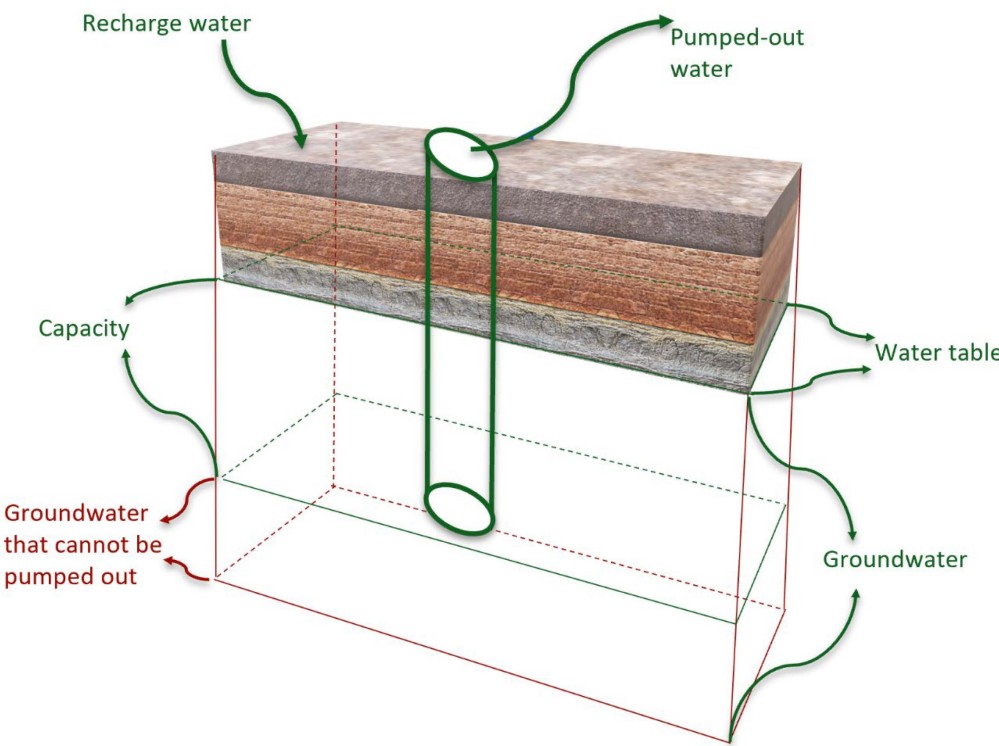

**Figure 1.** Conceptual plot illustrating the capacity of a well (Constraints (3) and (4)). The volume of groundwater below the deepest point of the well cannot be pumped out for drinking or irrigation.

*Deterministic MIP problem reformulation:*

The constraints (3) and (4) can be written into $\sum_{j=1}^{J} y_{ij} \le s * \left( w_j x_j - l_{swl_j} x_j \right)$, where $w_j x_j$ is a bilinear term. To linearize the bilinear term, we reformulate the problem using McCormick Envelope, which is a relaxation technique for bilinear non-convex nonlinear

programming problems [25]. The reformulation can be performed by replacing the bilinear term into a new variable (letting $w_j x_j = v_j$) and adding four sets of constraints. Below are the reformulated problems:

$$\min_x \sum_{j=1}^J c_j x_j + \sum_{i=1}^I \sum_{j=1}^J t_{ij} y_{ij} + \sum_{j=1}^J u_j w_j, \tag{10}$$

$$s.t. \qquad \sum_{j=1}^J y_{ij} = d_i, \ \ \forall i \tag{11}$$

$$\sum_{j=1}^J y_{ij} \le s(v_j - l_{swl_j} x_j), \ \ \forall i \tag{12}$$

$$v_j \ge \left(l_{swl_j} + l_{min}\right) x_j, \ \ \forall j \tag{13}$$

$$v_j \ge w_j + w_{max} x_j - w_{max}, \ \ \forall j \tag{14}$$

$$v_j \le w_j + \left(l_{swl_j} + l_{min}\right) x_j - \left(l_{swl_j} + l_{min}\right), \ \ \forall j \tag{15}$$

$$v_j \le w_{max} x_j, \ \ \forall j \tag{16}$$

$$\sum_{i=1}^I \sum_{j=1}^J y_{ij} \le r \tag{17}$$

$$w_j \le w_{max}, \ \ \forall j \tag{18}$$

$$w_j \ge \left(l_{swl_j} + l_{min}\right) x_j, \ \ \forall j \tag{19}$$

$$y_{ij} \ge 0, \ \ \forall i, \ \forall j \tag{20}$$

$$x_j \in \{0, \ 1\}, \ \forall j \tag{21}$$

2.1.2. Two-Stage Stochastic Mixed Integer Programming Model

Two-stage mixed-integer programming is an advanced optimization technique used to make decisions under uncertainty. In the first stage, decisions are made based on initial information before the uncertainty is revealed. In the second stage, additional decisions are adjusted after the uncertainty is known, allowing the model to respond dynamically to new information [26]. This approach is particularly useful for complex systems where future conditions are uncertain. In this section, the sets, parameters, decision variables, formulation, and reformulation of the two-stage SMIP are described.

*Sets:*

- $I$: set of farm sites, indexed by $i$.
- $J$: set of potential well sites, indexed by $j$.
- $M$: set of farm demand scenarios, indexed by $m$.

*Parameters:*

- $c_j$: construction fixed cost at site $j$.
- $u_j$: unit cost for drilling one meter at site $j$.
- $p_m$: probability of the demand scenario $m$.
- $d_{im}$: demand at farm $i$ for scenario $m$.
- $t_{ij}$: transportation cost from $j$ to $i$.
- $l_{swl_j}$: static water level of the well at site $j$.
- $r$: average recharge quantity
- $s$: area of the study region
- $w_{max}$: the deepest depth the well can be drilled.
- $l_{min}$ : the minimum difference between the depth of the well $w_j$ and static water level $l_{swl_j}$.

*Decision variables:*

- $x_j$: binary, equal to 1 if the well at location j is placed, otherwise 0.
- $y_{ijm}$: quantity of water that is transported from site *j* to farm *i* for demand scenario m.
- $w_j$: depth of the well to drill at site *j*.
- $K_j$: capacity of the well at site *j*.

*Two-stage SMIP problem formulation:*

We formulate the problem into a two-stage stochastic integer programming problem. The first stage decision variables are $x_j$ denoting whether a well at location *j* is placed, $w_j$, denoting depth of the well to drill at site *j*, and $K_j$ denoting the capacity of the well at site *j*. The second stage decision variable is $y_{ij}$, denoting the quantity of water that is transported from site *j* to farm *i* for demand scenario m. For the first stage, the objective is to minimize the total fixed construction costs and total drilling depth costs plus the expected cost of the second stage decision on transportation costs.

The first-stage constraints (23), (24), (26), and (27) are identical to the constraints in the deterministic mixed-integer programming model (constraints (6), (7), (8), and (9), respectively). Constraint (25) defines $K_j$ as the volume of groundwater above the deepest point of the well. The second-stage constraints are also similar to their corresponding constraints in the deterministic model, with the key difference being the incorporation of uncertain demand scenarios *m*.

Specifically, constraint (29) ensures that the demand for all scenarios is satisfied. Constraint (30) requires that each constructed well *j* has a capacity $K_j$, and the total quantity of water transported from well *j* to all farms for each demand scenario m must be less than or equal to $K_j$. Constraint (31) mandates that the total pumped-out water for each demand scenario m must be less than the quantity of recharged water. Finally, constraint (32) is the standard non-negativity constraint. The two-stage stochastic integer program is defined as

$$\min_x \sum_{j=1}^{J} c_j x_j + \sum_{j=1}^{J} u_j w_j + E_m Q(x,\ m), \tag{22}$$

$$s.t. \quad w_j \leq w_{max}, \quad \forall j \tag{23}$$

$$w_j \geq \left(l_{swlj} + l_{min}\right) x_j, \quad \forall j \tag{24}$$

$$K_j = s * \left(w_j - l_{swl_j}\right), \quad \forall j \tag{25}$$

$$w_j \geq 0, \quad \forall j \tag{26}$$

$$x_j \in \{0,\ 1\}, \quad \forall j \tag{27}$$

where $E_m$ denote mathematical expectation with respect to m. *Q(x, m)* is the value of the second-stage for a given realization of the random vector m.

$$Q(x,\ m) = \min \sum_{m=1}^{M} \left(p_m \left(\sum_{i=1}^{I} \sum_{j=1}^{J} t_{ij} y_{ijm}\right)\right), \tag{28}$$

$$s.t. \quad \sum_{j=1}^{J} y_{ijm} = d_{im}, \quad \forall i, \forall m \tag{29}$$

$$\sum_{j=1}^{J} y_{ijm} \leq K_j x_j, \quad \forall i, \forall m \tag{30}$$

$$\sum_{i=1}^{I} \sum_{j=1}^{J} y_{ijm} \leq r, \quad \forall m \tag{31}$$

$$y_{ij} \geq 0, \quad \forall i, \forall j \tag{32}$$

***Two-stage SMIP reformulation and deterministic equivalent form:***

The constraints (25) and (30) can be written into $\sum_{j=1}^{J} y_{ijm} \leq s * \left( w_j x_j - l_{swl_j} x_j \right)$, where $w_j\, x_j$ is a bilinear term. To linearize the bilinear term, we reformulate the problem using McCormick Envolope, which is a relaxation technique for bilinear non-convex nonlinear programming problems. The reformulation can be performed by replacing the bilinear term into a new variable (letting $w_j x_j = v_j$) and adding four sets of constraints. The objective is to minimize the summation of the total fixed cost for construction, the total transportation cost, and the cost for digging different depths of wells. We reformulate and define the problem in a deterministic equivalent form:

$$min \sum_{j=1}^{J} c_j x_j + \sum_{j=1}^{J} u_j w_j + \sum_{m=1}^{M} \left( p_m \left( \sum_{i=1}^{I} \sum_{j=1}^{J} t_{ij} y_{ijm} \right) \right), \tag{33}$$

$$s.t. \qquad \sum_{j=1}^{J} y_{ijm} = d_{im}, \quad \forall i, \forall m \tag{34}$$

$$\sum_{j=1}^{J} y_{ijm} \leq s(v_j - l_{swl_j} x_j), \quad \forall j, \forall m \tag{35}$$

$$v_j \geq \left( l_{swl_j} + l_{min} \right) x_j, \quad \forall j \tag{36}$$

$$v_j \geq w_j + w_{max} x_j - w_{max}, \quad \forall j \tag{37}$$

$$v_j \leq w_j + \left( l_{swl_j} + l_{min} \right) x_j - \left( l_{swl_j} + l_{min} \right), \quad \forall j \tag{38}$$

$$v_j \leq w_{max} x_j, \quad \forall j \tag{39}$$

$$\sum_{i=1}^{I} \sum_{j=1}^{J} y_{ijm} \leq r, \quad \forall m \tag{40}$$

$$w_j \leq w_{max}, \quad \forall j \tag{41}$$

$$w_j \geq \left( l_{swl_j} + l_{min} \right) x_j, \quad \forall j \tag{42}$$

$$y_{ij} \geq 0, \quad \forall i, \forall j \tag{43}$$

$$x_j \in \{0, 1\}, \quad \forall j \tag{44}$$

### 2.2. Study Area

From the regional point of view, the study area is located in the northeastern part of the Omo Basin, southern Main Ethiopian Rift, in the western flank of the Bilate watershed (Figure 2a,b). The Bilate watershed spans a latitude of $6°34'$ to $8°6'$ N and a longitude of $37°46'$ to $38°18'$ E, with a total area of 5276.25 km$^2$. The study area in the western flank of the Bilate watershed spans a latitude of $7°17'$ to $7°23'$ N and a longitude of $37°46'$ to $37°49'$ E, covering approximately 43.6 km$^2$. Based on the Digital Elevation Model (DEM), the elevation ranges from 2255 m to 2803 m (Figure 2c). This study area is characterized by high elevation and belongs to a semi-arid climate zone.

The diverse geological features of the Bilate watershed further influence the movement and availability of groundwater in this region. The geological features of the upper Bilate watershed are quite varied, playing a major role in shaping the area's underground water sources. This region is mainly made up of rocks dating back to the Cenozoic era, particularly basalts from the Oligocene to Miocene periods, which form the geological foundation. In the Rift Valley and the highlands are rhyolites and trachyte (acidic volcanic rocks) resulting from recent Quaternary volcanic activities, which add to the intricate geological composition of the area. Moreover, a considerable number of sedimentary deposits, such as lacustrine sediments, are vital for storing and facilitating the movement of groundwater resources [27]. The geological formations in the area significantly affect how groundwater moves due to faults and fractures. These structures can impede groundwater flow depending on how

permeable they are. One distinctive characteristic of the Rift Valley is its graben zones running from northeast to southwest. These zones were shaped by forces, and they are filled with a mix of volcanic and sedimentary materials [28]. These geological formations have implications for groundwater systems. The mix of sedimentary rocks forms an aquifer network where fractured volcanic rocks act as efficient aquifers because of their high permeability, while sedimentary layers can serve as both aquifers and aquitards. Faults and fractures play a role in groundwater replenishment and release by creating pathways for water to flow through the layers [27].

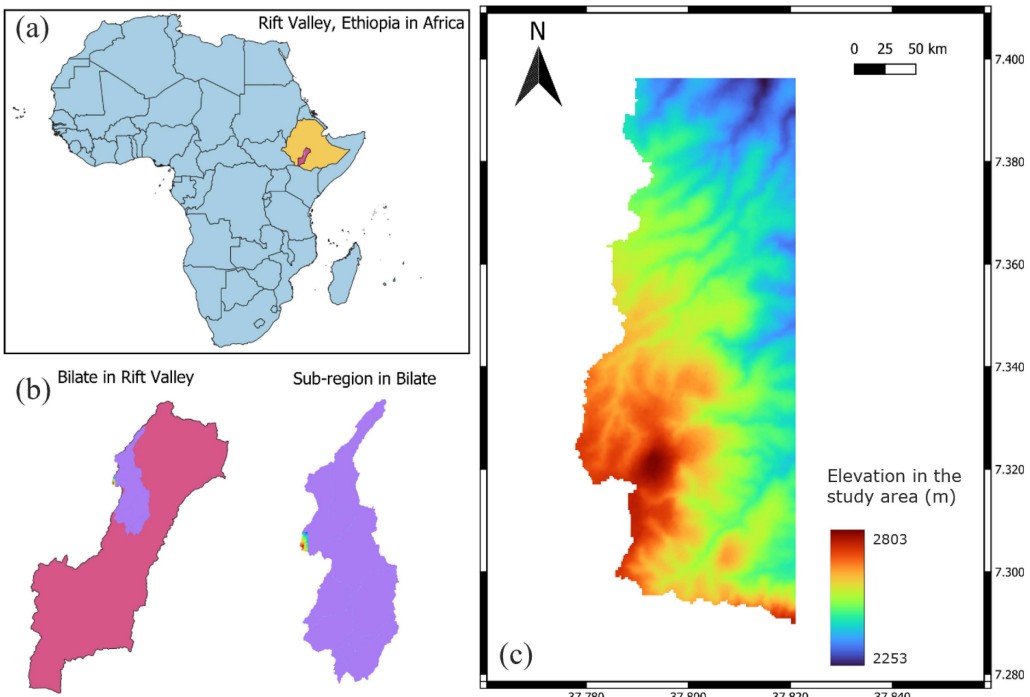

**Figure 2.** (**a**) Location of the southern part of the Main Ethiopian Rift in Ethiopia, Africa; (**b**) Bilate watershed (violet polygon) within the Southern Main Ethiopian Rift Valley (left) and the location of the study area within the Bilate watershed (right); (**c**) Simplified digital elevation model (DEM) of the area of interest (on the western edge of the Bilate watershed).

To determine the farm locations for the optimization model, a grid with a resolution of 1 km by 1 km was created. The center points of each grid cell were then generated and designated as farm locations, resulting in a total of 43 farms. This is for purposes of demonstration; in actual application, we would want data on where the farms really are. Previously, we created a grid with a resolution of 100 m by 100 m to predict groundwater levels within the Bilate region [19]. These 100 m resolution grid points are considered potential well locations, totaling 4264 potential sites. Figure 3 illustrates the farm and well locations on the map.

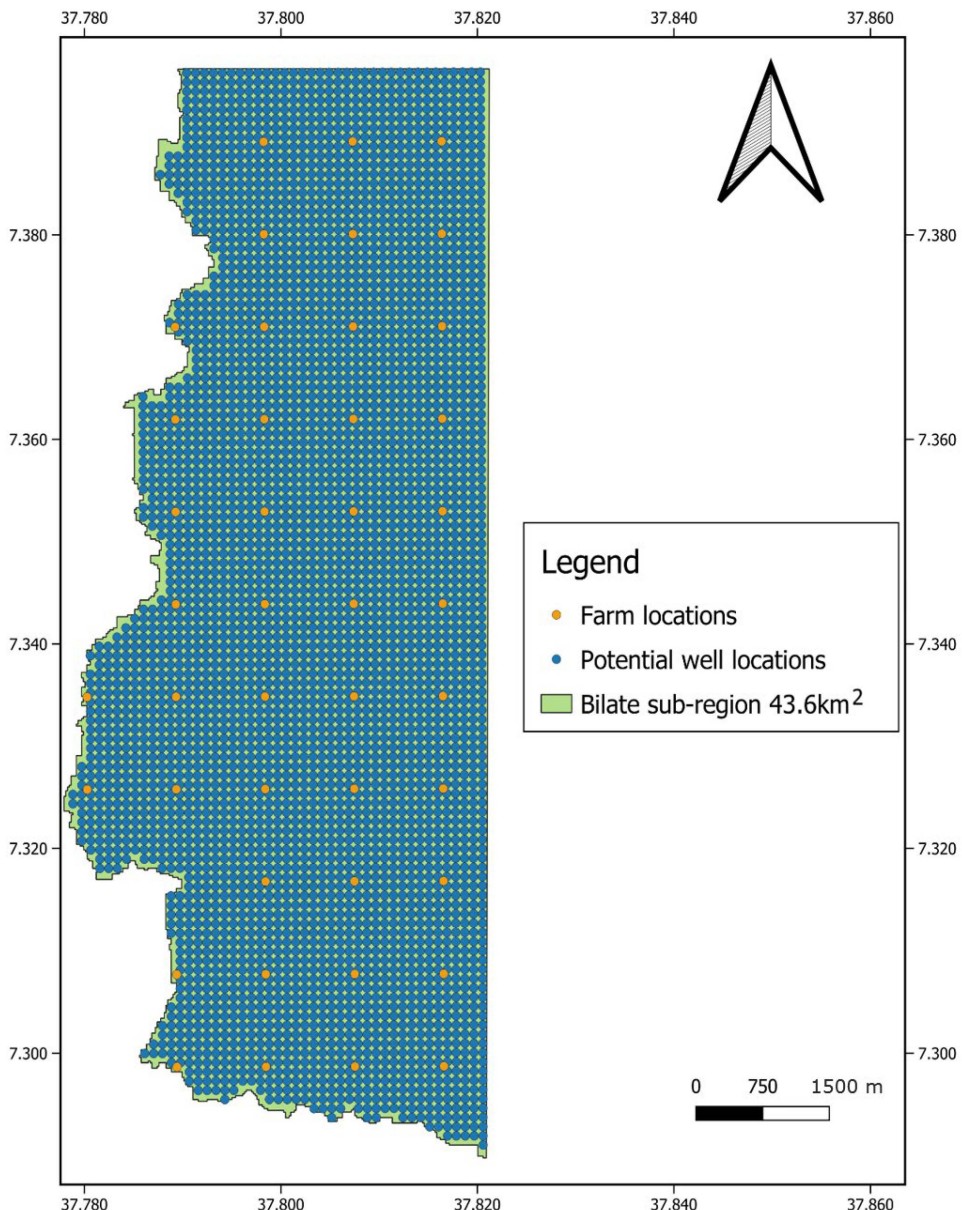

**Figure 3.** Farm and potential well locations in the study area (western flank of Bilate watershed).

*2.3. Data*

In this section, the initial settings of all the parameters are introduced. These settings were determined in consultation with Ethiopian hydrology experts who have extensive field experience.

### 2.3.1. Construction Costs and Per-Meter Drilling Costs

In this study, we initially assume a fixed construction cost of 5000 USD for each well site and a unit cost of 100 USD per meter drilled to solve the optimization problems. To ensure the robustness of our findings, we conduct a sensitivity analysis by considering a range of fixed cost values. The details and results of this sensitivity analysis will be presented in the next section.

### 2.3.2. Transportation Costs

We consider two scenarios for calculating the transportation costs from well site $j$ to farm $i$. One is that the farm elevation is greater than the elevation of the well site, so that water needs to be pumped uphill. In this case, we assume that the transportation costs are

very large, set at 9999 USD, when the pipe length between the farm and well is greater than 1000 m or the elevation difference between the farm and well is greater than 200 m. For the case where the pipe length is less than 1000 m and the elevation difference is less than 200 m, we use the formula below to calculate the transportation costs:

We consider two scenarios for calculating the transportation costs from well site j to farm *i*. In the first scenario, the farm elevation is greater than the elevation of the well site, requiring water to be pumped uphill. In this case, we assume that the transportation cost is prohibitively high, set at 9999 USD, if the pipe length between the farm and well exceeds 1000 m or if the elevation difference between the farm and well exceeds 200 m. In the other case, where the pipe length is less than 1000 m and the elevation difference is less than 200 m, we calculate the transportation costs using the following formula:

$$t_{ij} = u_c * E = u_c * \left( H_{ij} + h_{f1} * L_{ij} \right), \text{ for } H_{ij} <= 200 \text{ and } L_{ij} <= 1000$$

where $u_c$ is the unit cost, $E$ is the total energy, $H$ is the elevation difference between the farm and well, $h_{f1}$ is the friction loss for PVC pipe for scenario 1, and L is the pipe length.

The second scenario considers the situation where the farm elevation is lower than the elevation of the well, allowing water to flow downhill by gravity. In this case, we assume there is no cost for pumping. However, if the pipe length exceeds 1000 m, we assume a very high transportation cost. If the pipe length is less than 1000 m, the transportation costs are calculated using the following formula:

$$t_{ij} = u_c * E = u_c * h_{f2} * L_{ij}, \text{ for } L_{ij} \leq 1000$$

where $u_c$ is the unit cost, $E$ is the total energy, $h_{f2}$ is the friction loss for PVC pipe for scenario 2 and $L$ is the distance (pipe length).

To estimate the friction loss, the Hazen–Williams equation is used for this study. The equation is defined as:

$$h_f = \frac{10.67}{d^{4.8704}} \left( \frac{Q}{C} \right)^{1.85}$$

where $h_f$ denotes the friction head loss in meters over the water pipe, d is the inside diameter in meters, Q is the flow rate in cubic meters per second, and C is the roughness coefficient. For PVC pipe, the standard C value is 150 [29]. We assume a 3-inch PVC pipe is used, giving d = 0.0762 m. The flow rate Q is assumed to be 280 gallons per minute (gpm) for scenario 1, and 140 gpm for scenario 2 [30]. Converting these flow rates to cubic meters per second, we have approximately Q = 0.0177 $m^3$/s for scenario 1 and Q = 0.0088 $m^3$/s for scenario 2. Plugging these values into the equation, we obtain a friction loss of 0.1603 m for scenario 1 and 0.0445 m for scenario 2.

### 2.3.3. Water Demand

For the deterministic mixed-integer programming problem, a single demand scenario (m = 1) has been considered, with the demand set to, for example, 1000 kg/km$^2$ per year. This simplifies the problem by assuming a constant demand across the entire study area. In this study, deterministic MIP models are constructed using demand values ranging from 800 to 1200, with increments of 100 (Section 3.1). These demand values were recommended by local water resource management experts based on regional water usage patterns and anticipated demand.

In contrast, the two-stage stochastic mixed-integer programming problem considers multiple demand scenarios (m = 10), which are assumed to follow a uniform distribution, each with a probability of 0.1. This approach accounts for the uncertainty in demand, allowing for a more robust optimization model that can adapt to various possible demand levels. To enhance the robustness and reliability of the model, different distribution assumptions are explored in the sensitivity analysis. This helps to understand the impact of varying

demand patterns on the optimization outcomes and ensures that the proposed solutions are effective under a range of possible future scenarios.

### 2.3.4. Average Groundwater Recharge Quantity

The groundwater recharge has been estimated in a previous study within this dissertation using an observation-constrained land surface model (LSM). This model provides straightforward recharge estimates for the data-limited, water-scarce Rift Valley basin in Ethiopia. It primarily utilizes publicly available data from NASA's GLDAS Noah model, covering the years 2000 to 2022, offering a cost-effective method for estimating groundwater recharge over large regions. To validate this approach, we conducted a comprehensive analysis that integrates groundwater recharge, precipitation, and land cover and land use data, alongside temporal and spatial variability assessments. We also compared our results with existing literature on similar geographical and climatic regions. The findings suggest that our model aligns well with other methods of estimating groundwater recharge. According to the results, the study area used for the optimization model has an approximate yearly average recharge quantity of 323,000 $kg/km^2$.

### 2.3.5. Static Water Level

The static water level for each potential well location has been predicted in a previous study within this dissertation using machine learning algorithms. Models including multiple linear regression, multivariate adaptive regression splines, artificial neural networks, random forest regression, and gradient boosting regression (GBR) have been developed to forecast the static water level using 75 borehole data in Bilate watershed, a sub-basin in Rift Valley. The study incorporated 20 independent variables, such as elevation, soil type, and seasonal data for precipitation, specific humidity, wind speed, day and night land surface temperatures, and the Normalized Difference Vegetation Index (NDVI). Among these models, GBR demonstrated the highest performance, achieving an average R-squared value of 0.77 and a median absolute error of 19 m on the testing data. The static water level for the potential well locations in the optimization model has been predicted using the best-performing GBR model. The predicted static water levels in this sub-region range from 60 to 137 m.

### 2.3.6. Other Data

To formulate the optimization problem, the area of the study region (s) is required. As previously described, the study region covers 43.6 $km^2$. According to Kebede, groundwater depths are categorized as follows: very shallow (0–30 m), shallow (30–100 m), deep (100–250 m), and very deep (over 250 m). Our study focuses on finding shallow groundwater for irrigation. However, the results from the GBR model show that the predicted static water levels range from 1.6 to 245.9 m in the Bilate watershed and from 60 to 137 m within the study area for the optimization model. Considering the cost per meter of drilling, the maximum depth for well drilling ($w_{max}$) in this optimization model is set at 140 m. Additionally, the minimum difference between the depth of the well and the static water level ($l_{min}$) is set to be 1 m.

## 3. Results

In this study, the Gurobi solver was used to address the optimization problems, ensuring efficient and accurate solutions [31]. The software tools employed included Python 3.12 for scripting and algorithm implementation [32] and QGIS 3.24.1 for geographic data processing and visualization [33]. These tools provided a robust framework for developing and analyzing the well placement optimization problem.

### 3.1. Model Optimization Results

Three models were employed in this study: a deterministic mixed-integer programming (MIP) model with demand values set at 800, 900, 1000, 1100, and 1200; a two-stage

stochastic mixed-integer programming (SMIP) model with demand scenarios randomly generated from different uniform distributions (U (400, 1200), U (500, 1300), U (600, 1400), U (700, 1500), and U (800, 1600)); and another two-stage SMIP model with demand scenarios randomly generated from five different normal distributions with means of 800, 900, 1000, 1100, and 1200, each with a standard deviation of 300. The means of the demand scenarios generated from each uniform distribution are 800, 900, 1000, 1100, and 1200, which match the settings of the deterministic MIP model. A similar rationale applies to the two-stage SMIP model with the normal distribution. The results show that the average total costs for the three models were 42,264,421, 42,298,103, and 42,317,641, respectively (Table 1). The demand values considered in this study range from 800 to 1200 kg/km$^2$, with increments of 100. As mentioned in Section 2.3.3, these values were recommended by local water resource management experts, reflecting regional water usage patterns and anticipated demand. Considering different scenarios allows us to account for the inherent uncertainty in demand patterns, which can significantly influence well placement and resource allocation decisions.

**Table 1.** Results for the optimization models.

| Deterministic MIP with Fixed Demand | | | | | | |
|---|---|---|---|---|---|---|
| Demand (kg/km$^2$) | 800 | 900 | 1000 | 1100 | 1200 | Average |
| Optimized number of wells | 43 | 44 | 45 | 46 | 46 | 45 |
| Total costs (USD) | 42,177,807 | 42,224,273 | 42,265,578 | 42,306,933 | 42,347,516 | 42,264,421 |
| Total fixed construction costs (USD) | 215,000 | 220,000 | 225,000 | 230,000 | 230,000 | 224,000 |
| Total drilling costs (USD) | 41,719,176 | 41,728,938 | 41,738,701 | 41,748,463 | 41,758,326 | 41,738,721 |
| Total transportation costs (USD) | 243,631 | 275,335 | 301,877 | 328,470 | 359,190 | 301,700 |
| Two-stage SMIP (Demand scenarios: uniform distribution) | | | | | | |
| Demand Instance | U (400, 1200) | U (500, 1300) | U (600, 1400) | U (700, 1500) | U (800, 1600) | Average |
| Optimized number of wells | 45 | 45 | 46 | 46 | 46 | 45.6 |
| Total costs (USD) | 42,215,743 | 42,256,368 | 42,297,266 | 42,339,950 | 42,381,187 | 42,298,103 |
| Total fixed construction costs (USD) | 225,000 | 225,000 | 230,000 | 230,000 | 230,000 | 228,000 |
| Total drilling costs (USD) | 41,748,893 | 41,758,756 | 41,768,351 | 41,778,241 | 41,787,793 | 41,768,406 |
| Total transportation costs (USD) | 241,850 | 272,612 | 298,915 | 331,709 | 363,394 | 301,696 |
| Two-stage SMIP (Demand scenarios: normal distribution) | | | | | | |
| Demand Instance | N (800, 300) | N (900, 300) | N (1000, 300) | N (1100, 300) | N (1200, 300) | Average |
| Optimized number of wells | 44 | 44 | 46 | 46 | 46 | 45.2 |
| Total costs (USD) | 42,233,743 | 42,275,352 | 42,316,817 | 42,360,022 | 42,402,269 | 42,317,641 |
| Total fixed construction costs (USD) | 220,000 | 220,000 | 230,000 | 230,000 | 230,000 | 226,000 |
| Total drilling costs (USD) | 41,762,470 | 41,772,055 | 41,783,204 | 41,793,020 | 41,802,142 | 41,782,578 |
| Total transportation costs (USD) | 251,273 | 283,297 | 303,613 | 337,002 | 370,127 | 309,062 |

### 3.2. Optimal Well Layouts

As demand values increased in the deterministic MIP model, the number of wells placed also increased, leading to higher total costs. This is because higher demand requires additional groundwater extraction to meet the increased water needs. As a result, more wells are necessary to ensure sufficient water supply across the region. For uniform distribution demand scenarios, slightly more wells were placed, but the total costs were lower than those from models with demand scenarios generated from a normal distribution, possibly due to deeper wells being placed, leading to higher drilling costs. Table 1 summarizes the optimized number of wells, total costs, and cost components for the three models.

Maps were created to illustrate the optimized well placement for each experiment and each model (Figures 4–6). In the deterministic MIP model experiments, well placements/layouts were largely similar, with differences observed in the placement of wells 419, 420, 506, 3666, 3780, and 3781 across experiments based on assumed water demand. For the two-stage SMIP model using different uniform distributions, wells 276, 419, 420,

3551, 3666, 3781, 3629, and 3744 were placed slightly differently across experiments. With an increase in the mean demand, the number of wells placed slightly increased. In the two-stage SMIP model using a normal distribution, decisions regarding the placement of wells 338, 419, 420, 506, 1457, 1458, 2170, 2398, 2285, 3551, 3629, 3666, 3744, 3780, and 3781 varied slightly across models.

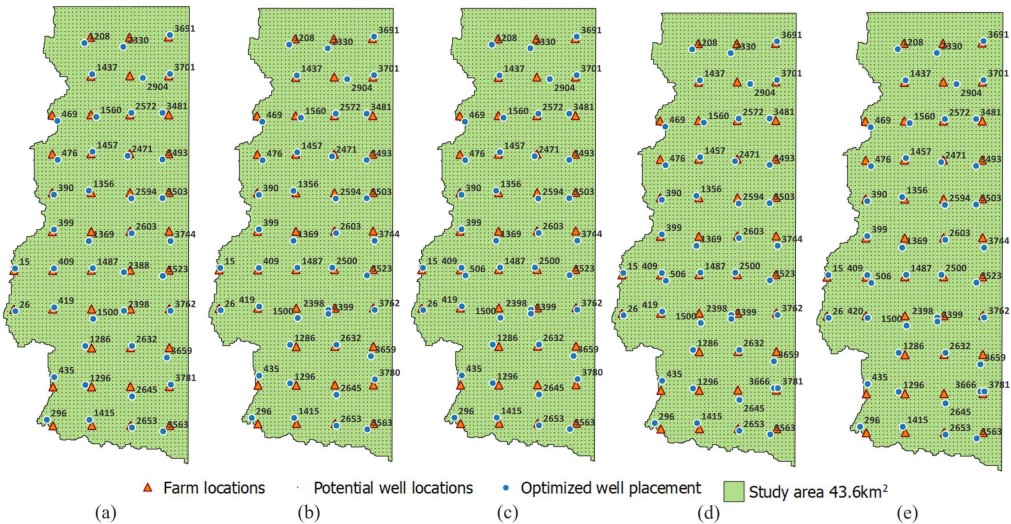

**Figure 4.** (**a**–**e**) Five deterministic MIP experiments assuming water demands of 800, 900, 1000, 1100, and 1200 kg/km², respectively. (Note: The numbers on the map represent the well ID). The layout of a few wells is different across different models, including well ID 419, 420, 506, 3666, 3780, and 3781, which are all located in the southern part of the study area.

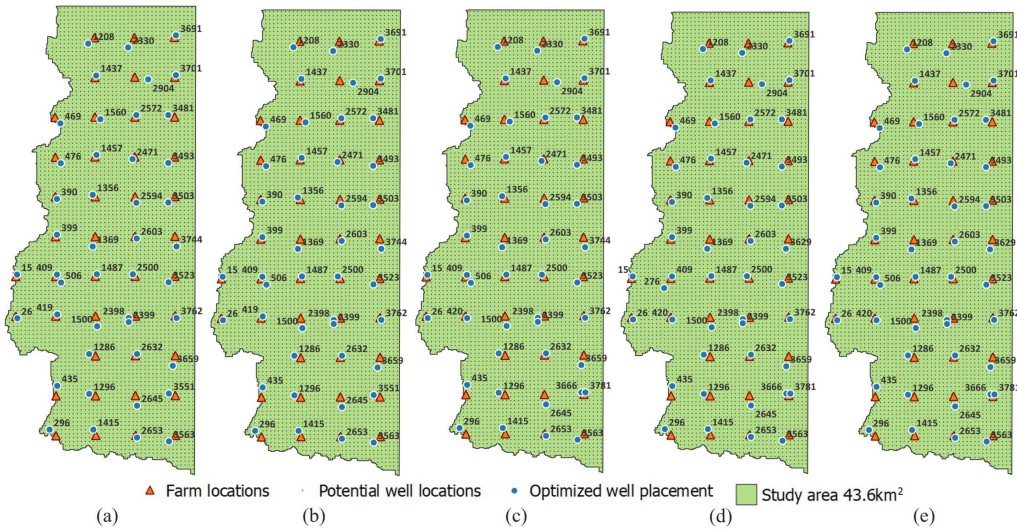

**Figure 5.** (**a**–**e**) Illustration of five two-stage SMIP experiments depicting water demand scenarios randomly generated from five uniform distributions: U (400, 1200), U (500, 1300), U (600, 1400), U (700, 1500), and U (800, 1600), respectively. (Note: The numbers on the map represent the well ID). The layout of a few wells is different across different models, including well ID 276, 419, 420, 3551, 3666, 3781, 3629, and 3744. Well 276 is located in the eastern part of the study area in map (**d**). Wells 3629 and 3744 are situated in the mid-eastern region, while the remaining wells are located in the southern part of the study area.

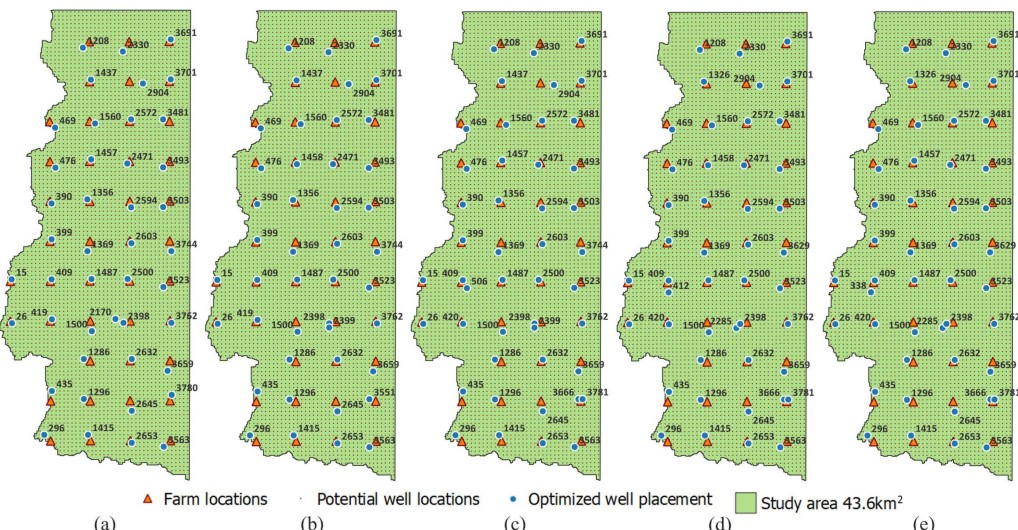

**Figure 6.** (**a**–**e**) Visualization of five two-stage SMIP experiments depicting water demand scenarios randomly generated from five normal distributions: N (800, 300), N (900, 300), N (1000, 300), N (1100, 300), and N (1200, 300), respectively. (Note: The numbers on the map represent the well ID). The layout of a few wells is different across different models, including well ID 338, 419, 420, 506, 1457, 1458, 2170, 2398, 2285, 3551, 3629, 3666, 3744, 3780, and 3781. Well 338 is located in the eastern part of the study area in map (**e**). Wells 1457 and 1458 are located in the northwestern part. Wells 3629 and 3744 are situated in the mid-eastern region, while the remaining wells are located in the southern part of the study area.

## 4. Discussion

### 4.1. Impact of Different Parameters

In this study, we consider the experiment for the uniform distribution U (600, 1400) for the demand scenarios of the two-stage SMIP as the base model. The impact of different parameters demonstrated in this section is created based on this model.

#### 4.1.1. Impact of Different Fixed Construction Costs

The fixed construction costs are the coefficients of the decision variable $x_j$ (whether to place a well at location $j$). The objective total costs include three components: total fixed construction costs, total per-meter drilling costs (also known as unit costs), and total transportation costs. The range of fixed construction costs is set from 3000 to 18,000, with an increment of 1000. As the fixed construction costs increase (Figure 7), the total costs also rise.

The differences between consecutive objective values are calculated and plotted in Figure 8. From this plot, we can see that when the fixed cost increases from 5000 to 6000, the difference between the objective values is 46,000. When the fixed cost increases from 6000 to 7000, the difference is approximately 44,000. This clear drop from 46,000 to 44,000 occurs because, when the fixed cost increases from 6000 to 7000, the total number of wells placed decreases from 46 to 45. Additionally, the well layout changes, indicating that the model chooses to drill deeper wells with higher drilling costs rather than maintaining the same number of wells with higher construction costs to minimize the total costs. Another observation is an increase in total cost from 44,000 to 44,700. This occurs because, when the fixed costs increase from 7000 to 8000, the number of wells decreases from 45 to 44. Consequently, the well layout changes to satisfy the water demand, leading to approximately 7600 higher transportation costs and 100 higher per-meter drilling costs at the 8000 fixed cost compared to the 7000 fixed cost. Additionally, there is a notable drop in total cost from 44,000 to 41,454. This decrease happens because, when the fixed costs increase from 11,000 to 12,000, the number of wells to be placed decreases from 44 to 43. Subsequently, the number of wells placed remains constant at 43, resulting in

a consistent difference of 43,000 in objective values as the fixed construction cost increases by increments of 1000.

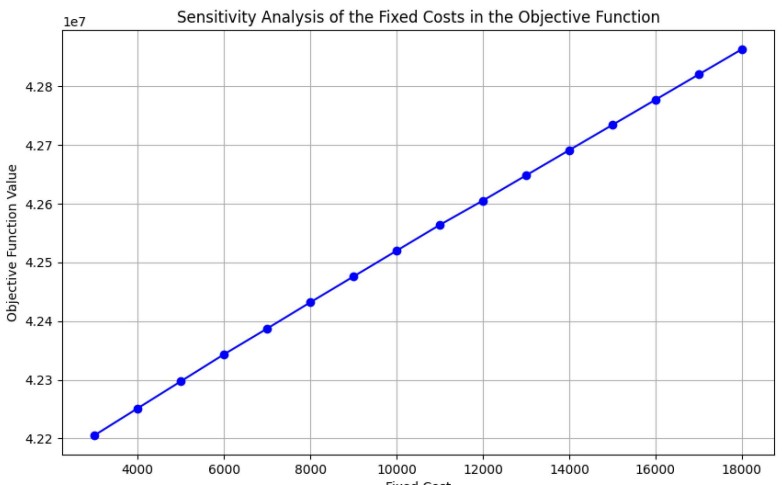

**Figure 7.** Impact of the fixed construction costs on the objective values.

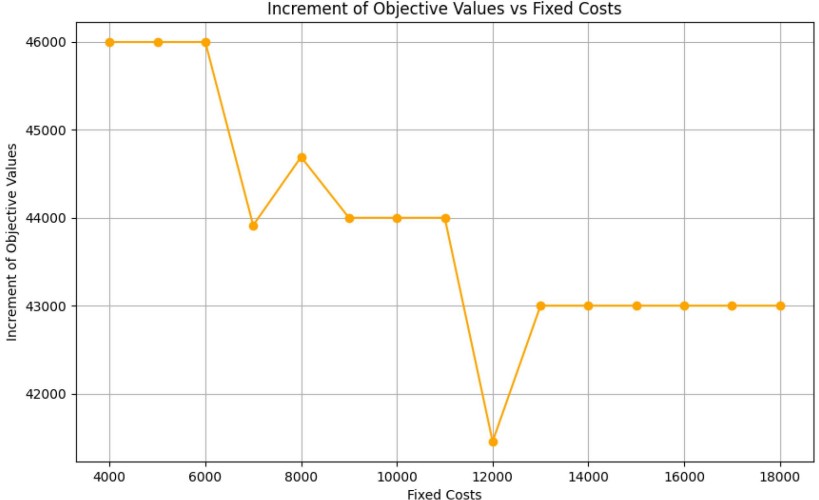

**Figure 8.** Increment of objective values vs. fixed construction costs.

### 4.1.2. Impact of Different Per-Meter Drilling Costs

The per-meter drilling costs are the coefficient of the decision variable $w_j$ (depth of the well placed at location j). Similarly, the range of the per-meter drilling costs is set from 30 to 150, with an increment of 10. As the per-meter drilling costs increase, the total costs also rise (Figure 9).

Figure 10 shows the differences between consecutive objective values as the per-meter drilling cost increases. From this plot, we can see a clear drop in the increment of the objective when the per-meter drilling cost increases from 110 to 120. This occurs because, as the per-meter drilling cost becomes higher, the model tends to select shallower wells that are slightly farther from the farm to minimize the total costs.

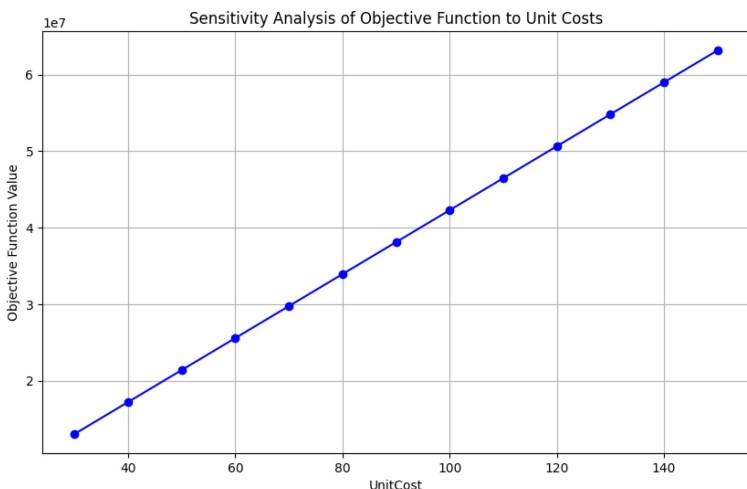

**Figure 9.** Impact of the different per-meter drilling costs on the objective values.

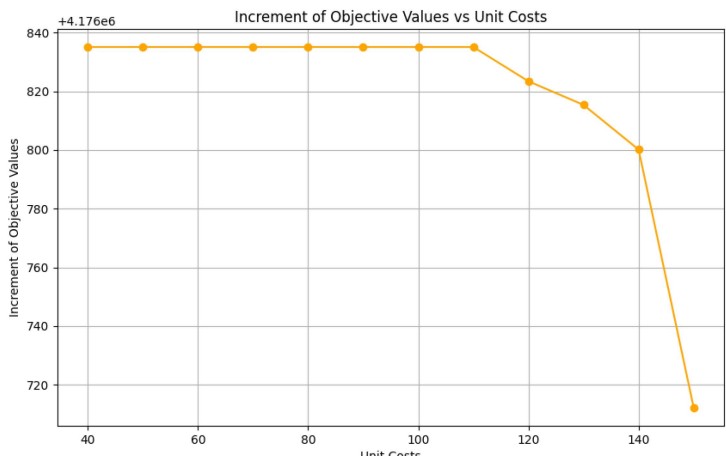

**Figure 10.** Increment of objective values vs. per-meter drilling costs (unit costs).

4.1.3. Impact of Different Demand Scenarios

Since the demand scenarios are generated randomly, the increments in the objective values between each pair of demand scenarios are not constant. Some increments are big and some are small. Two big increment examples are described next.

First, in Figure 11b, there is an obvious increase in the objective values when moving from U (1200, 2000) to U (1300, 2100). This can be explained by the fact that as the mean demand increases from 1600 to 1700, the number of open wells increases from 47 to 49. This increase in the number of open wells is the main contributor to the substantial increment in the objective values (Figure 12a).

When increasing the demand range from U (600, 1400) to U (1100, 1900), the number of open wells remains the same (Figure 12a). However, the objective values see a greater increase when moving from U (800, 1600) to U (900, 1700); see Figure 11a. This phenomenon can be explained by the fact that as the mean demand increases, different well layouts are likely to be required to meet the rising demand. This leads to higher transportation costs and necessitates deeper wells, which increases capacity and results in higher total drilling costs. Note that the capacity directly impacts the total quantity of water transported from the well to the farm, which in turn affects the demand (Figure 12).

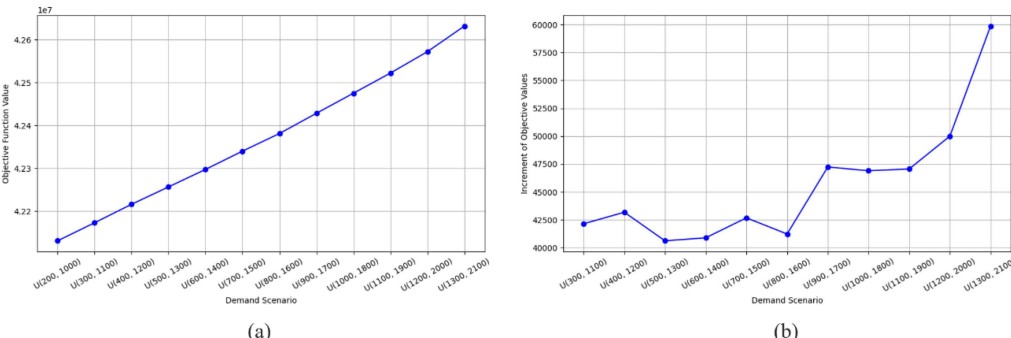

**Figure 11.** (**a**) Objective values vs. different demand scenarios (**b**) Increment of the objective values vs. different demand scenarios.

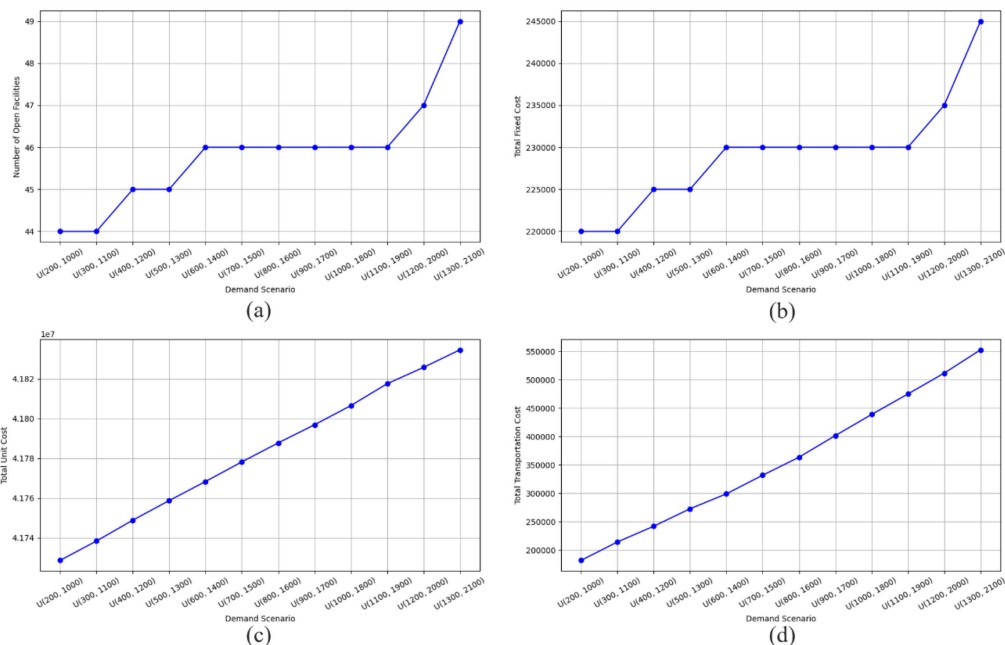

**Figure 12.** (**a**) Number of open wells vs. demand scenarios; (**b**) Total fixed costs vs. demand scenarios; (**c**) Total unit costs vs. demand scenarios; (**d**) Total transportation costs vs. demand scenarios.

### 4.2. Out-of-Sample Performance

To evaluate out-of-sample performance, we first formulate the problem with a deterministic demand (in-sample demand data) and solve it. The solution of the first-stage decision variables, whether to place a well at site $j$ ($x_j$) and the depth of the well to drill at site $j$ ($w_j$), is then fixed. Using these fixed first-stage decision variables, we solve a new problem with stochastic demand scenarios (out-of-sample demand data) to obtain the objective values. By fixing the same first-stage decision variable results, the second-stage demand scenarios can be randomly generated from the same distribution multiple times, producing a set of objective values.

Next, we formulate the problem with a stochastic demand scenario (in-sample demand data) and solve it. The solutions of the first-stage decision variables are then fixed to solve a problem with the same stochastic demand scenario (out-of-sample demand), but with newly randomly generated demand scenarios, iterating this process multiple times to obtain objective values. Finally, the two cases—the deterministic case and the stochastic case—can be compared by analyzing the mean and standard deviation of their objective values to evaluate out-of-sample performance.

In this study, four cases were investigated, each iterated 50 times. The uniform distribution was chosen to range from 600 to 1400, with a mean of 1000. For the normal

distribution, a mean of 1000 and a standard deviation of 155 were selected. This choice was made to ensure that 99% of the values fall between 600 and 1400 in a normal distribution, as the standard deviation is approximately calculated as 400/2.576 = 155. The deterministic demand was set at 1100. This value was chosen instead of 1000 to ensure the feasibility of the model.

It is important to note that the number of wells to place is 45 for the deterministic MIP with a demand of 1000. In contrast, the SMIP with demand scenarios generated from a uniform distribution U (600, 1400) indicates that 46 wells should be placed. Therefore, if we use an in-sample demand with a deterministic value of 1000, the placement of 45 wells may not satisfy the out-of-sample demand scenarios generated from U (600, 1400).

The mean and standard deviation of the objective values from experiments using the out-of-sample demand data for the four cases are summarized in Table 2. We compared Case 1 with Case 2 and Case 3 with Case 4, and created boxplots in Figures 13 and 14. The results indicate that the deterministic case 1 achieves 11% higher average out-of-sample objective costs compared to the stochastic case 2. Additionally, the average costs for deterministic case 3 are 4% higher than those for stochastic case 4. This demonstrates that the stochastic approach yields lower average costs and is more effective than the deterministic approach. The boxplots also show that the stochastic cases perform better than the deterministic cases, with tighter distributions and lower variability in the objective values. This indicates that stochastic models consistently perform better across a wider range of out-of-sample scenarios.

**Table 2.** Results of the out-of-sample performance.

|  | **Case 1** | **Case 2** | **Case 3** | **Case 4** |
|---|---|---|---|---|
| **In-sample demand** | Deterministic with demand = 1100 | Uniform distribution U (600, 1400) | Deterministic with demand = 1100 | Normal distribution N (1000, 155) |
| **Out-of-sample demand** | Uniform distribution U (600, 1400) | Uniform distribution U (600, 1400) | Normal distribution N (1000, 155) | Normal distribution N (1000, 155) |
| **Mean** | 47,124,183 | 42,494,243 | 44,200,128 | 42,625,237 |
| **Standard deviation** | 938,970 | 107,608 | 437,315 | 172,968 |

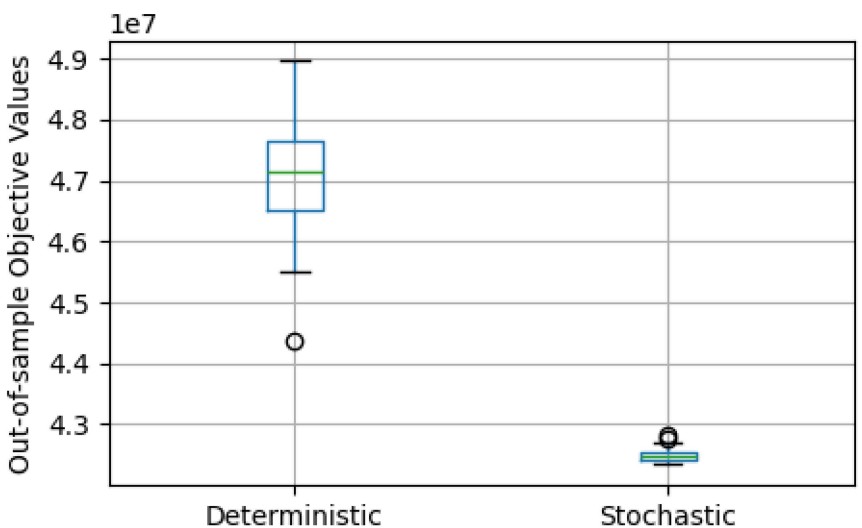

**Figure 13.** Comparison of the out-of-sample performance between a deterministic in-sample demand and a stochastic out-of-sample demand generated from a U (600, 1400).

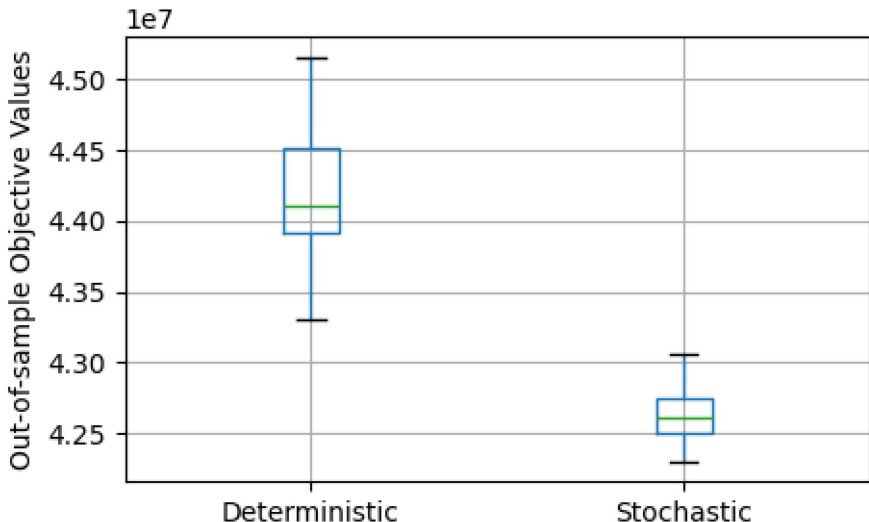

**Figure 14.** Comparison of the out-of-sample performance between a deterministic in-sample demand and a stochastic out-of-sample demand generated from a N (1000, 155).

These results emphasize that incorporating demand uncertainty through stochastic optimization (Case 2 and Case 4) enables more adaptive and responsive well placement strategies, reducing the risk of inefficiencies or resource shortages that may occur with deterministic models (Case 1 and Case 3), which tend to result in higher objective costs. Stochastic models are better equipped to handle variability, offering consistent performance across various scenarios, which is essential for long-term groundwater management. Their flexibility and resilience to changing demand provide a more robust strategy for well placement and resource allocation, ensuring the sustainable management of groundwater resources.

### 4.3. Limitations and Future Research Directions

The optimization study has several limitations that should be acknowledged. Firstly, the model is restricted to a very small region due to computational constraints, which limits the generalizability of the findings to larger areas. This restriction means that the results may not fully capture the complexities and variances that could occur in a broader geographical context. However, it is important to recognize that applying these algorithms to larger regions or areas with greater hydrological variability presents its own set of challenges, even though the stochastic model shows clear advantages for well placement optimization. Secondly, the study does not incorporate data on the actual locations of farms, which is crucial for accurate well placement and optimizing irrigation strategies. Without this data, the model's applicability to real-world scenarios is limited. Additionally, while the study examines the uncertainty of demand scenarios, it does not consider other important uncertainties, such as variations in static water levels and groundwater recharge. These factors could significantly impact the effectiveness of the well placement optimization and should be included in future research to provide a more comprehensive analysis.

Future research could focus on conducting extensive scenario analyses to explore the impacts of different demand scenarios on well placement optimization, which could provide valuable insights into the robustness and reliability of the model under various geological and hydrogeological conditions. Examining the effect of uncertainty in static water levels and groundwater recharge should be considered, also with regard to the progressing climate change. Additionally, exploring and comparing different optimization methods could provide a broader perspective on the most effective approaches for well placement optimization. This development could enable optimizing well placement over a larger region, potentially providing a broader impact. Furthermore, collecting data on the actual locations of farms is crucial for more accurate well placement. Another future research direction could be developing adaptive optimization techniques that can dynami-

cally update well placement decisions based on real-time data and changing conditions, which could enhance the flexibility and responsiveness of the decision support system.

## 5. Conclusions

In this study, we formulated both deterministic MIP and two-stage SMIP problems to optimize well placement with minimal total costs while satisfying farm demand and ensuring sustainable groundwater extraction. Our study area is located in the northeastern part of the Omo Basin in the southern Main Ethiopian Rift, on the western flank of the Bilate watershed. Experiments were conducted with different distributions of demand scenarios for the two-stage SMIP models. The results show that the average total costs for the MIP are slightly higher than those for the SMIP models. This is intuitive because the two-stage stochastic model incorporates the benefit of long-term planning by considering uncertain demand scenarios. By accounting for these uncertainties, the SMIP can optimize well placement more effectively over time, potentially avoiding costly misallocations and adjustments that a deterministic MIP model might incur when it does not account for demand variability. Consequently, the SMIP's more comprehensive approach can lead to more cost-effective solutions in the long run.

Maps of well placement for each experiment have been created, revealing slight changes in the layout of the wells near a few farms in the south across different experiments with varying demand scenarios. In practice, this could guide drilling decisions by incorporating expert judgment based on soil conditions and geographic features. The analysis of the impact of varying fixed construction costs, per-meter drilling costs, and demand scenarios on the objective values shows that total costs increase with these costs and demand. The increments in objective values relative to these three parameters have also been examined and explained. The results of the out-of-sample performance show that the stochastic cases perform better than the deterministic cases, with tighter distributions and lower variability in the objective values (11% and 4% lower costs than deterministic cases 1 and 3, respectively). This indicates that the solutions from the stochastic models are more flexible and resilient to changes in demand.

**Author Contributions:** Conceptualization, W.L. and K.B.L.; methodology, W.L.; software, W.L.; validation, W.L., M.M.F., K.B.L., P.H., R.D.-B. and K.V.; formal analysis, W.L.; investigation, W.L. and K.B.L.; resources, M.M.F., R.D.-B. and K.V.; data curation, W.L.; writing—original draft preparation, W.L.; writing—review and editing, W.L., M.M.F., K.B.L., P.H., R.D.-B. and K.V.; visualization, W.L.; supervision, K.B.L., P.H. and R.D.-B.; project administration, K.B.L., P.H., and R.D.-B.; funding acquisition, K.B.L., P.H., R.D.-B. and K.V. All authors have read and agreed to the published version of the manuscript.

**Funding:** This research was partially supported by a graduate research fellowship to W.L. from Czech Geological Survey development aid project No. ET-2023-006-RO-43040 and George Mason University's Center for Resilient and Sustainable Communities.

**Data Availability Statement:** The data presented in this study will be available on interested request from the corresponding author.

**Acknowledgments:** This work represents a collaboration among George Mason University, Arba Minch University, and Global Map Aid with support by the Czech Geological Survey. This elaboration was processed as a part of the development aid project by the Czech Geological Survey No. ET-2023-006-RO-43040 (to K. Verner) entitled "Improving the quality of life by ensuring availability and sustainable management of water resources in Sidama Region and Gamo and Gofa Zones (Ethiopia)" financed by the Czech Republic through the Czech Development Agency.

**Conflicts of Interest:** Author Rupert Douglas-Bate was employed by the company Global MapAid. The remaining authors declare that the research was conducted in the absence of any commercial or financial relationships that could be construed as a potential conflict of interest.

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
