# Peer review of "Optimizing Well Placement for Sustainable Irrigation: A Two-Stage Stochastic Mixed Integer Programming Approach"

_water, doi:10.3390/w16192715_

Round 1
Reviewer 1 Report
Comments and Suggestions for Authors
I have carefully read the manuscript titled “Optimizing Well Placement for Sustainable Irrigation: A Two-Stage Stochastic Mixed Integer Programming Approach” by Li et al. The manuscript is in good shape. The authors provided some useful information about the development of two-stage stochastic mixed integer programming (SMIP) models to optimize well placement. However, there are still some improvements:
1.In the last three paragraphs of “Introduction” section, the author try to concise the objection of this research. However, it should be concise. I suggest the authors to rewrote it.
2.The section of “2.1. Problem Formulation and Reformulation” is suggested to be deleted.
3. More importantly, the hydrological geology should be provided in details, which is the base of conceptual model.
4. The author consider different scenarios. However, why? No reason? The details should be provided.
5. The conclusion should be rewrote.
Author Response
Thank you deeply for your thorough feedback and insights. Please find our responses to your comments in the attached document. Thanks.

Reviewer 2 Report
Comments and Suggestions for Authors
Optimizing Well Placement for Sustainable Irrigation: A Two- Stage Stochastic Mixed Integer Programming Approach by Wanru Li, Mekuanent Muluneh Finsa, Kathryn Blackmond Laskey, Paul Houser, Rupert Douglas-Bate and Kryštof Verner
This study aims to model and optimize optimal drilling locations to minimize costs while meeting farm demand and ensuring sustainable groundwater development. To achieve this objective, deterministic mixed-integer programming (MIP) and two-stage stochastic mixed-integer programming models are constructed and compared, using a case study in the western flank of the Bilate watershed (westernmost part of the Sidama Region) in southern Ethiopia.
I recommend that the manuscript be accepted for publication with minor revision.
General comments
1. I feel as though the paper has adequately described the approach as applied, and the results, apart from two comments as follows.
2. Maintaining water resource management objectives under imperfect knowledge usually requires risk management approaches, including adaptive management. As information on future demand or other factors (e.g. recharge and water table behaviour) becomes known, management and policies can change accordingly. Often, the characteristics of the groundwater system only become known after extraction has occurred for some time. The knowledge of optimal locations of pumping in response to uncertainty in demand can inform the risk management strategy. However, the introduction has not provided the context for doing this. Rather, it avoided these issues, as it leapt from unlocking the potential of groundwater resources into strategic decision-making in well drilling and placement and then quickly into optimisation of location being explained as a holistic approach to water management. I would suggest that most groundwater-sourced irrigation areas do not have bore layouts that are optimal in the sense described here; and that for most readers, more context is required.
3. The limitations in optimisation algorithms in considering larger areas, incorporating data on farm locations; uncertainties in recharge and static water level and applicability in different hydrogeological conditions would mean that such algorithms are far from routine. However, it would be useful to understand how uncertainty in demand might affect optimal solutions through describing the learnings from the model in much more detail. I feel as though some more interpretation of the results would be much more useful. Even the results are not included in the conclusions. The benefits, as described, would seem minor (negative impact initially, changing to 4-11% less cost.
4. The discussion of future research is best not included in the conclusions, or at least in the detail in the manuscript but in discussion.
Author Response
Thank you so much for recommending our manuscript to be accepted for publication with minor revisions. Please find our responses to your comments in the attached document. Thank you!

Reviewer 3 Report
Comments and Suggestions for Authors
Very interesting paper on irrigation wells systems and its optimization. However interesting approach is presented generally clear, for readers not familiar with the topic it could be not sound if presented experimental optimum location of irrigation wells vary from another - fig 4, 5 and 6 looks the same and it is very hard to distinguish which wells location differ from another.
Author Response
We deeply appreciate your insightful feedback. Please find our responses to your comments in the attached document. Thank you.
